# Characterization of Almond Scion/Rootstock Communication in Cultivar and Rootstock Tissues through an RNA-Seq Approach

**DOI:** 10.3390/plants12244166

**Published:** 2023-12-15

**Authors:** Álvaro Montesinos, María José Rubio-Cabetas, Jérôme Grimplet

**Affiliations:** 1Centro de Biotecnología y Genómica de Plantas, Universidad Politécnica de Madrid—Instituto Nacional de Investigación y Tecnología Agraria y Alimentación (UPM-INIA/CSIC), 28223 Madrid, Spain; al.montesinos@externos.upm.es; 2Centro de Investigación y Tecnología Agroalimentaria de Aragón (CITA), Departamento de Ciencia Vegetal, Gobierno de Aragón, Avda. Montañana 930, 50059 Zaragoza, Spain; mjrubioc@cita-aragon.es; 3Instituto Agroalimentario de Aragón-IA2 (CITA-Universidad de Zaragoza), Calle Miguel Servet 4 177, 50013 Zaragoza, Spain

**Keywords:** *Prunus dulcis*, transcriptome, tree architecture, root development, scion/rootstock interaction

## Abstract

The rootstock genotype plays a crucial role in determining various aspects of scion development, including the scion three-dimensional structure, or tree architecture. Consequently, rootstock choice is a pivotal factor in the establishment of new almond (*Prunus amygdalus* (L.) *Batsch*, syn *P. dulcis* (Mill.)) intensive planting systems, demanding cultivars that can adapt to distinct requirements of vigor and shape. Nevertheless, considering the capacity of the rootstock genotype to influence scion development, it is likely that the scion genotype reciprocally affects rootstock performance. In the context of this study, we conducted a transcriptomic analysis of the scion/rootstock interaction in young almond trees, with a specific focus on elucidating the scion impact on the rootstock molecular response. Two commercial almond cultivars were grafted onto two hybrid rootstocks, thereby generating four distinct combinations. Through RNA-Seq analysis, we discerned that indeed, the scion genotype exerts an influence on the rootstock expression profile. This influence manifests through the modulation of genes associated with hormonal regulation, cell division, root development, and light signaling. This intricate interplay between scion and rootstock communication plays a pivotal role in the development of both scion and rootstock, underscoring the critical importance of a correct choice when establishing new almond orchards.

## 1. Introduction

In modern orchards, rootstocks serve a dual purpose of selecting specific root system traits and conferring desirable agronomic traits to trees and fruits [1,2]. These effects on scion development have been extensively documented across various tree species, ranging from tree vigor to yield or fruit quality [1,3,4,5,6]. Vigor and architecture in particular are important for high-density planting systems. Recently, molecular approaches have been conducted in woody plant species to elucidate the underlying mechanisms at the molecular level [7,8]. However, the analysis of the scion effect on the rootstock has been limited to the graft formation, examining the processes occurring during the vascular union, which results in vascular regeneration and the establishment of the graft junction [9,10]. However, little is known about how the scion can influence the phenotypic traits exhibited by the rootstock, including nutrient assimilation, pathogen resistance, and root development [11]. These traits may be influenced in a variable manner depending on the specific scion cultivar that is grafted onto the rootstock.

Rootstock development is regulated by various phytohormones, participating in the regulation of cell elongation, cell division, and cell differentiation [12,13]. Like the aerial part of the plant, auxin plays a vital role in governing various processes in roots, including root patterning, cell division, and cell elongation [14,15,16]. Strigolactones (SLs) act in conjunction with auxin, exerting control over lateral root formation and root-hair elongation, while mediating root responses to environmental changes [17,18,19]. Cytokinins (CKs) promote cell differentiation and cell division in different root tissues and counteract auxin by inhibiting lateral root formation [20,21,22,23]. Gibberellic acid (GA) contributes to the maintenance of root cell proliferation and cell elongation in the meristem, while suppressing lateral root formation [24,25,26]. Brassinosteroids (BRs) play a crucial role in regulating root meristem activity and also participate in the regulation of lateral root initiation and root cell elongation [27,28]. Ethylene (ET) modulates meristem maintenance, promoting cell division and acting in opposition to auxin in lateral root formation [29,30].

Light signaling exerts significant control over plant development through diverse mechanisms. In plants, the circadian clock regulates various developmental processes in response to fluctuations in exposure to light, including seed germination, hypocotyl elongation, root growth, and flowering [31,32]. The regulation of the circadian clock is also intertwined with carbohydrate metabolism and nutrient assimilation [33]. Furthermore, the shade avoidance response plays a role in regulating plant growth, which depends on the ratio of red light to far-red light (R:FR) detected by phytochrome photoreceptors phyA and phyB. Alterations in this ratio elicit changes in the auxin flux, thereby influencing the direction and activity of plant growth [34,35,36,37,38].

In this study, we carried out a transcriptomic analysis of the responses associated with both the rootstock influence on the scion and the scion influence on the rootstock. To investigate these interactions, we grafted two almond commercial cultivars characterized by contrasting architecture and vigor traits onto two almond × peach (*Prunus amygdalus* (L.) Batsch, syn *P. dulcis* (Mill.). × *P. persica* (L.) Batsch). This resulted in a total of four combinations. Our goal was to elucidate the biological processes and molecular responses that were specifically affected in both the upper and lower regions of the graft site.

## 2. Results

### 2.1. ‘Isabelona’ and ‘Lauranne’ Vigor Was Influenced by the Rootstock

Data on tree architecture were obtained for the four graft combinations, ‘Isabelona’/Garnem^®^, ‘Isabelona’/‘GN-8’, ‘Lauranne’/Garnem^®^, and ‘Lauranne’/‘GN-8’ (Figure 1). As the trees were still in their early stages and had not yet developed branches, measurements were restricted to trunk length (Length) and the diameters of both the scion (d_Scion) and the rootstock (d_Rootstock). Due to the inherent challenges associated with accurately measuring root architecture, no data were collected on the root architecture.

Combinations involving ‘Lauranne’ as the scion displayed higher Length values (Table 1, Figure 1). Regarding the rootstocks, the use of the vigor-conferring rootstock Garnem^®^ resulted in higher length values for both cultivars, presenting higher values than when grafted onto the dwarfing rootstock ‘GN-8’ (Table 1, Figure 1).

Trunk diameter (d_Scion), often represented as Trunk Cross Sectional Area (TCSA), is commonly used as a measure of tree vigor. Similar to the length values, ‘Lauranne’ exhibited higher d_Scion values than ‘Isabelona’. Additionally, when grafted onto Garnem^®^, both cultivars displayed higher d_Scion values compared to grafting onto ‘GN-8’ (Table 1). However, no significant difference was observed in the diameters of the rootstocks (d_Rootstock), although the mean values were slightly lower when ‘Isabelona’ was used as the scion (Table 1).

### 2.2. Rootstock Only Influenced Gene Expression in Combinations with ‘Isabelona’

A transcriptomic analysis from cultivar samples was carried out to verify whether their gene expression was affected by the rootstock genotype during the early stages of development (Appendix A). A Principal Component Analysis (PCA) was conducted using gene expression data from the four combinations, with each gene serving as a variable. The first component (PC1; 83.5%) accounted for most of the variability (Figure 2). The samples were primarily arranged by their respective cultivars, while the rootstock genotype had a minimal impact on their distribution. Notably, no DEGs were detected for the ‘Lauranne’ combinations. In contrast, 221 DEGs were identified in the combinations involving ‘Isabelona’ as the scion (Appendix A).

Combinations of ‘Isabelona’ grafted onto the vigor-conferring rootstock Garnem^®^ resulted in notable alterations in gene expression levels associated with auxin regulation, predominantly through repressive mechanisms. Furthermore, distinct expression patterns were observed for genes involved in stress hormone signaling, specifically ABA and ET pathways. Moreover, DEGs related to plant development, cell wall reorganization, and the regulation of growth-related hormones GA and BRs were identified (Table 2).

### 2.3. Scion/Rootstock Interaction in Almond Affected Rootstock Molecular Profile

The cultivar effect of commercial almond cultivars ‘Lauranne’ and ‘Isabelona’ on the rootstock transcriptome was analyzed in a vigorous rootstock like Garnem^®^, and a dwarfing rootstock such as ‘GN-8’. We carried out a PCA using the expression of each gene as variables for the four different scion/rootstock combinations. The first two components explained 51.5% of the variability, while none of the other variables explained more than 14%. PC1 and PC2 explained 28.8% and 22.7% of the variability, respectively. In the PCA, there was a clear separation between the four different combinations (Figure 3A). Combinations with Garnem^®^ as rootstock are in the lower-left corner while combinations with ‘GN-8’ are in the upper-right corner. Therefore, there is a clear effect of the rootstock, and it can be observed in the gene expression, with individuals clearly segregating depending on which scion, ‘Lauranne’ or ‘Isabelona’, is grafted onto them (Figure 3A). A total of 36 genes were only differentially expressed in combinations with ‘GN-8’ as rootstock, while 342 DEGs were identified in combinations with Garnem^®^. Both rootstocks shared 28 DEGs (Figure 3B). Therefore, while the number of DEGs was considerably more elevated when comparing combinations that had been grafted onto the vigor-conferring rootstock Garnem^®^, than those grafted onto the dwarfing rootstock ‘GN-8’, around half of DEGs found in the later were shared between both rootstocks.

### 2.4. Garnem^®^ Transcriptome Is More Affected by Cultivar Effect Than the ‘GN-8’ Transcriptome

The rootstock exhibited varying degrees of cultivar influence. In the case of ‘GN-8’, a limited number of DEGs were observed, indicating a relatively minor impact on gene expression. Conversely, the rootstock Garnem^®^ displayed a higher degree of differential expression, suggesting a more pronounced influence on the transcriptome (Figure 3B). Among the limited number of genes that showed differential expression in ‘GN-8’, several have homologues known to participate in plant development, such as *CLV1* (Prudul26A015064), *FIP37* (Prudul26A020498), *EF1A* (Prudul26A027600), and *VIM1* (Prudul26A008660). Furthermore, specific to root development, homologues of genes such as *IAMT1* (Prudul26A029624) and *DRMH3* (Prudul26A007496) were found to be less expressed in combinations where ‘Lauranne’ served as the cultivar (Table 3). Moreover, genes involved in hormonal response or transport, including homologues of *LAX3* (Prudul26A031524) and *RD22* (Prudul26A001972), also exhibited lower expression levels when ‘Lauranne’ was grafted onto the ‘GN-8’.

The impact of the cultivar on the transcriptome of the Garnem^®^ rootstock was found to be more pronounced, influencing multiple pathways associated with plant development. Several homologues of genes involved in auxin response, such as *IAA4* (Prudul26A024452) and *SAUR50* (Prudul26A025556), as well as auxin transport, like *LAX3* (Prudul26A031524), exhibited differential expression, with lower expression levels observed when ‘Lauranne’ was used as the scion (Table 4). Furthermore, genes involved in other hormonal regulatory pathways, including ABA, BR, CK, and ET response, as well as GA biosynthesis, were also identified as differentially expressed (Table 4). Specifically, homologues of genes related to ET response, such as *EGY3* (Prudul26A021691) and *ERF24* (Prudul26A019984), displayed increased expression in combinations where ‘Lauranne’ served as the scion. Conversely, homologues of BR response genes, including *TINY2* (Prudul26A010146) and *BAS1* (Prudul26A030744), exhibited lower expression levels in combinations with ‘Lauranne’ as the scion.

The cultivar genotype exerted a significant influence on growth-related processes, impacting cell division, cell wall reorganization, and organ development (Table 4). Homologues of genes involved in the cell division complex, such as *MCM* genes (Prudul26A029769, Prudul26A013222) and *RCC1* (Prudul26A003343, Prudul26A018495), displayed overexpression when ‘Lauranne’ was the cultivar. Similarly, homologues of genes with crucial roles in plant development, such as *GI* (Prudul26A016707) and *PAT1* (Prudul26A024652), were also found to be overexpressed when ‘Lauranne’ acted as the scion. Conversely, multiple other homologues of genes involved in various functions related to plant development, including *MAX1* (Prudul26A022418) and *FIP37* (Prudul26A020498), exhibited lower expression levels in combinations with ‘Lauranne’ as the cultivar. Notably, root development was also affected, with genes such as *AGL79* (Prudul26A020939) and *DRMH3* (Prudul26A007496) presenting reduced expression in combinations with ‘Lauranne’.

## 3. Discussion

Prior research has established that the rootstock genotype has the capacity to induce phenotypic alterations in the tree architecture of almond cultivars [39,40]. Nevertheless, the precise mechanisms underlying this intricate interplay between the scion and rootstock at the molecular level have yet to be comprehensively elucidated. In this study, we analyzed how two hybrid rootstocks (‘GN-8’ and Garnem^®^) modulate the transcriptome of two distinct almond commercial cultivars (‘Isabelona’ and ‘Lauranne’). Additionally, we explored how these cultivars might reciprocally influence the transcriptome of the respective rootstocks.

‘Isabelona’ has previously been described to exhibit reduced vigor and strong apical dominance, resulting in a phenotype characterized by limited branching and elongated trunks. Conversely, ‘Lauranne’ has been reported to display high vigor and weak apical dominance, leading to extensive branching and shorter trunks [39,40]. In the current study, the trees are in their initial year of growth, and they have not yet developed branches that compete with the main axis growth. Consequently, the increased vigor observed in the ‘Lauranne’ cultivar may contribute to the higher length values reported (Table 1). Ultimately, the observed differences in phenotype appear to predominantly hinge on the vigor displayed by each graft combination. However, it is plausible that the biological processes governing the development of specific tree architectures for each combination have already been established, despite not yet being visually discernible in these one-year-old plants.

The rootstock genotype had a restricted impact on the transcriptome of the studied cultivars (Figure 2; Appendix A). Whereas a limited number of DEG was discerned in combinations with ‘Isabelona’ as the cultivar, no DEGs were identified when ‘Lauranne’ was the scion. This observation reaffirms the limited malleability of these specific cultivars by the rootstock genotype, corroborating earlier findings [39]. This characteristic underscores the potential utility of these cultivars as an ideal platform for exploring whether the cultivar genotype can exert a modulatory effect on the expression profile of the rootstock.

In terms of the rootstock’s influence on ‘Isabelona’, various genes related to plant development displayed differential expression patterns (Table 2). Combinations in which Garnem^®^ served as the rootstock exhibited an upregulation of genes associated with auxin regulation, likely in a repressive manner [41,42,43,44], indicating a potential reduction in apical dominance [45,46]. Auxin-related genes such as Aux/IAA can be found in phloem and might actually be synthetized in the rootstock: Lower expression of IAA4 in persimmon induced vigor [47], which is consistent with our observation. As it appears downregulated in both the rootstock and the scion when Laurane is grafted with Garnem^®^ this RNA might be mobile. Additionally, DEGs involved in processes linked to active growth, such as cell proliferation and cell expansion [48,49,50], as well as genes promoting nitrogen assimilation [51,52,53], were upregulated in ‘Isabelona’/Garnem^®^ combinations. This upregulation likely contributed to the enhanced vigor observed in scions grafted onto Garnem^®^ (Figure 1; Table 1). Furthermore, genes that positively regulate stress responses and growth inhibition were upregulated when ‘GN-8’ was used as the rootstock [54,55,56], which may be linked to the limited cultivar growth caused by this dwarfing rootstock.

Given that distinct rootstocks have the capacity to influence almond scion architecture, potentially altering branch number and main axis growth, it is reasonable to consider that the grafted scion, in turn, exerts an influence on the rootstocks, triggering diverse mechanisms that could impact rootstock properties. This reciprocal effect has previously been documented in other plant species with respect to various traits, including the regulation of rootstock responses to low phosphorus (Pi) conditions and phloem sap metabolites [57,58]. In the present study, it was observed that the cultivar genotype did indeed influence the rootstock transcriptome (Figure 3). However, the extent of this influence was notably contingent on the specific rootstock employed. Although our samples were collected from the rootstock trunk, it is conceivable that the alterations in the transcriptome dynamics observed there may have broader ramifications for the entire root system.

The rootstock ‘GN-8’ exhibited a significantly lower number of DEGs compared to Garnem^®^ (Figure 3B), although approximately half of these DEGs were shared between the two rootstocks. Consequently, it appears that the cultivar effect on both rootstocks follows a similar direction, with the key distinguishing factor being the magnitude of this effect. The diminished intensity of the influence on the ‘GN-8’ transcriptome could potentially be attributed to the dwarfing characteristics associated with this particular rootstock. Such dwarfing rootstocks are often characterized by less active metabolic processes, which could render them less susceptible to modulation by the scion’s influence. This may elucidate the observed reduction in the extent of transcriptomic alteration in ‘GN-8’ as compared to Garnem^®^.

In both ‘GN-8’ and Garnem^®^ rootstocks, an evident transcriptomic profile emerged, favoring cell division and differentiation, and consequently, tree vigor, in graft combinations with ‘Lauranne’ as the scion (Table 3 and Table 4). This trend was manifested either through the upregulation of DEGs that promote cell division, root development, or cell wall formation [59,60,61,62,63,64,65,66,67,68,69], or the downregulation of genes that suppress plant development and lateral root formation [42,49,70,71,72,73,74,75,76]. Notably, genes associated with light perception and circadian clock regulation appeared to be influenced by the cultivar genotype (Table 4). Light availability exerts regulatory control over numerous plant developmental processes, with several pathways involved in growth regulation [77,78]. These pathways can influence lateral branch formation through mechanisms like shade avoidance [34,36]. Furthermore, the circadian clock, which is governed by light and other environmental cues, plays a pivotal role in regulating various aspects of plant development, including root growth [31,32,33]. While the genes associated with circadian clock regulation did not exhibit a discernible trend [79,80,81,82,83,84,85,86,87], it is apparent that these intricate processes can be impacted by the interaction between the scion and rootstock. This highlights the intricate interplay between scion and rootstock genotypes, which can influence processes related to plant development, including those regulated by light perception and the circadian clock.

## 4. Materials and Methods

### 4.1. Plant Material and Growth Conditions

For the experimental setup, two almond commercial cultivars, ‘Isabelona’ and ‘Lauranne’ were grafted onto two distinct hybrid rootstocks known as Garnem^®^ (a commercially available rootstock) and ‘GN-8’ (a newly selected accession), obtaining four different combinations. Both rootstocks are almond × peach (*P. amygdalus* (L.) Batsch, syn *P. dulcis* (Mill.). × *P. persica* (L.) Batsch) hybrid rootstocks. The selection of ‘Isabelona’ and ‘Lauranne’ cultivars was based on their reduced influence exerted by the rootstock on their apical dominance and branch formation phenotypes [39]. Grafted plants were supplied by the Agromillora Iberia S.L. (Barcelona, Spain) nursery in 2020. Subsequently, the plants were temporarily housed in a nursery before sample collection, which took place at the Centro de Investigación y Tecnología Agroalimentaria de Aragón (CITA). Standard nursery practices were maintained during the experimental period.

### 4.2. Phenotypic Data Collection in Nursery

Phenotypic data were collected for ten replicates of each of the four combinations, prior to sample collection. Three vigor-related parameters were measured: scion axis length (Length), scion trunk diameter (d_Scion), and rootstock trunk diameter (d_Rootstock). The length measurement was determined from the graft union. To quantify d_Scion and d_Rootstock, a caliper was employed to measure 20 mm above and 20 mm below the graft union, respectively.

### 4.3. RNA-Seq Analysis

During the summer of 2020, samples were collected from nine (three per sample) different individuals per combination, encompassing a region 50 mm below and above the graft union. RNA extraction was performed on these samples utilizing the CTAB method, as previously described, with modifications [88,89,90]. Stranded mRNA-Seq analysis was conducted at the Centro Nacional de Análisis Genómico (CNAG-CRG) in Barcelona, Spain. Sequencing was performed using the Illumina NovaSeq 6000 System, generating > 30 M PE reads per sample and a read length of 2 × 50 bp. The resulting FASTQ files were processed with FASTQ Groomer (Galaxy Version 1.1.1) [91]. Adapter sequences were subsequently removed from the reads using Trimmomatic (Galaxy Version 0.38.0) [91,92]. Alignment of the RNA-Seq data was performed using HISAT2 (Galaxy Version 2.2.1), employing a maximum intron length of 20,000 bp [93] and aligning to the *P. dulcis* ‘Texas’ Genome v2.0 [94]. Duplicated molecules were identified using the MarkDuplicates tool (Galaxy Version 2.18.2.2), and mate-pair information was confirmed using FixMateInformation (Galaxy Version 2.18.2.1) Picard tools (http://broadinstitute.github.io/picard, accessed on 30 November 2022). Gene expression was quantified using featureCounts (Galaxy Version 2.0.1+galaxy2) [95] using the *P. dulcis* ‘Texas’ Genome v2.0 gene annotation, which comprises 27,044 genes (https://www.rosaceae.org/analysis/295, accessed on 30 November 2022). Differential analysis of the count data was conducted using edgeR (Galaxy Version 3.36.0) with default settings [96], considering genes with a corrected *p*-value below 0.05 and a log2 fold change (log2FC) greater than 0.75 or less than −0.75 as differentially expressed. All procedures were performed using the Galaxy platform. Recent studies have demonstrated the robustness of RNA-Seq methods, with validation through qPCR confirming results for most transcripts. Discrepancies between the two techniques were observed in less than 2% of cases, primarily involving short and low-expressed genes [97,98]. As this data relates to Fragments Per Kilobase of transcript per Million (FPKM), differentially expressed genes were filtered to ensure they belonged to the top 98% of transcripts with the highest FPKM values.

### 4.4. Statistical Analysis

All statistical analyses were carried out in the R platform (https://cran.r-project.org/, accessed on 30 November 2022). Significant differences in phenotypic data were evaluated using an ANOVA test. These were assessed with a Tukey’s test (*p* < 0.05) using the agricolae R package (https://CRAN.R-project.org/package=agricolae, accessed on 30 November 2022). Principal component analysis (PCA) was conducted using R stats package with default parameters on the gene expression values for the all the genes in the four combinations.

## 5. Conclusions

The interaction between scion and rootstock in almond trees is bidirectional, exerting influence on the development of both the scion and the rootstock. In this study, we have identified several biological processes that undergo differential modulation depending on the rootstock genotype, impacting various aspects of cultivar growth and development. Additionally, this influence appears to produce a feedback mechanism within the rootstock developmental processes. Specifically, our findings demonstrate that cultivars displaying heightened vigor, exemplified by ‘Lauranne’, have a positive influence on root development. This positive impact enhances the radicular system’s capacity to efficiently absorb nutrients from the soil. Consequently, this nurtures scion growth, ultimately resulting in the robust and vigorous phenotype exhibited by ‘Lauranne’ when compared to ‘Isabelona’. Hence, the selection of an appropriate scion/rootstock combination is a pivotal determinant for the success of almond orchards. In intensive cultivation systems, the rootstock impact on tree vigor depends not only on its genotype but also on the complementary interactions with the scion. It is in this synergy that root development is optimized, consequently influencing overall tree growth and productivity.

## Figures and Tables

**Figure 1 plants-12-04166-f001:**
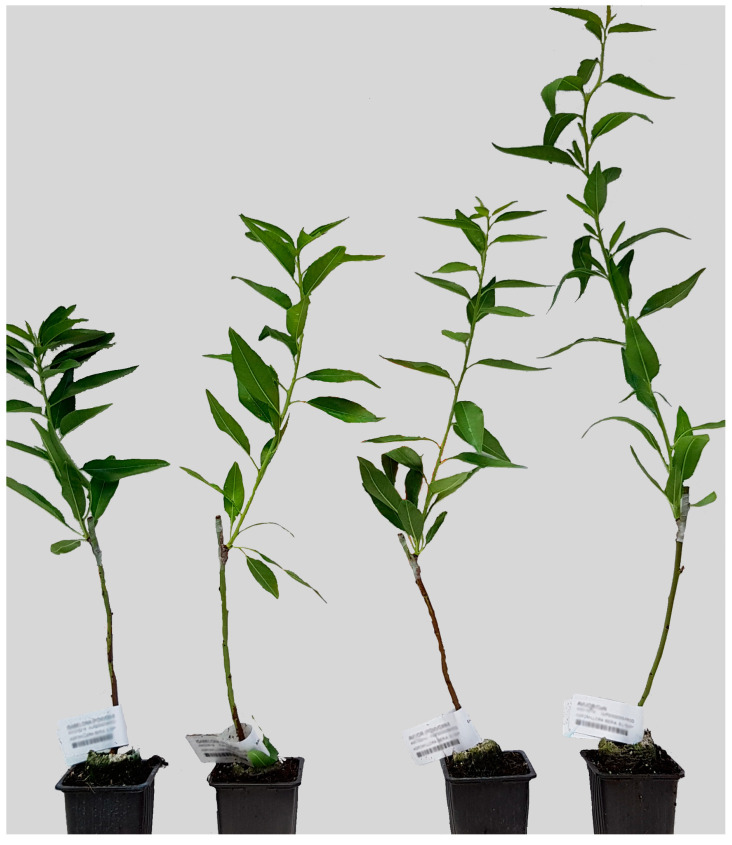
The scion/rootstock combinations showed differences in vigor response. From left to right: ‘Isabelona’/‘GN-8’, ‘Isabelona’/Garnem^®^, ‘Lauranne’/‘GN-8’ and ‘Lauranne’/Garnem^®^.

**Figure 2 plants-12-04166-f002:**
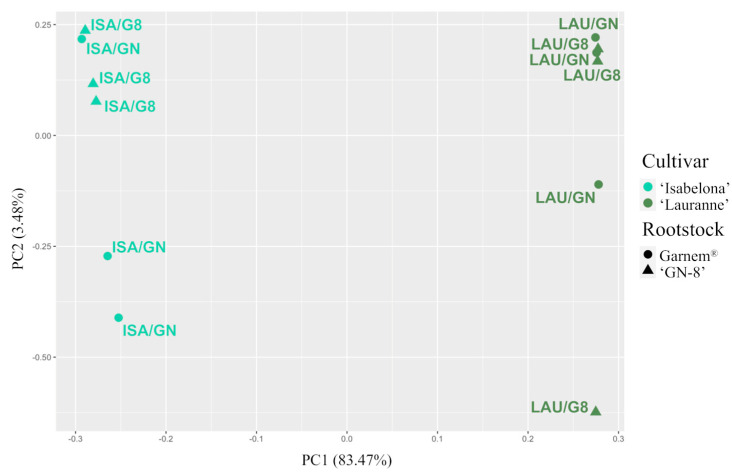
Principal component analysis (PCA) of the global expression profile data from cultivar samples of the four scion/rootstock combinations. ISA/GN: ‘Isabelona’/Garnem^®^; ISA/G8: ‘Isabelona’/‘GN-8’; LAU/GN: ‘Lauranne’/Garnem^®^; LAU/G8: ‘Lauranne’/‘GN-8’.

**Figure 3 plants-12-04166-f003:**
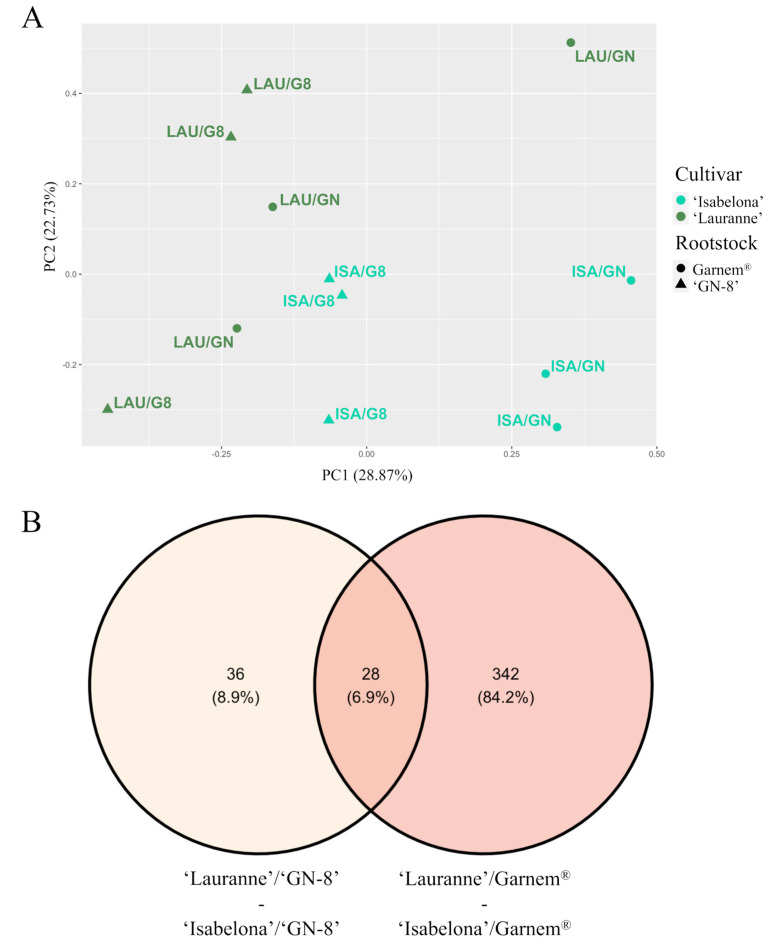
Analysis of the cultivar effect on the rootstock transcriptome data. (**A**) Principal component analysis (PCA) of the global expression profile data from rootstock samples of the four scion/rootstock combinations. ISA/GN: ‘Isabelona’/Garnem^®^; ISA/G8: ‘Isabelona’/‘GN-8’; LAU/GN: ‘Lauranne’/Garnem^®^; LAU/G8: ‘Lauranne’/‘GN-8’. (**B**) Venn diagrams of differentially expressed genes (DEGs) for the four scion/roostock combinations.

**Table 1 plants-12-04166-t001:** Analysis of architectural traits related to vigor in one-year-old scion/rootstock combinations.

Cultivar	Rootstock	Length (mm)	d_Scion (mm)	d_Rootstock (mm)
‘Isabelona’	‘GN-8’	210 a	2.63 a	4.25 a
	Garnem^®^	260 b	3.25 ab	4.36 a
‘Lauranne’	‘GN-8’	310 c	2.97 ab	4.50 a
	Garnem^®^	400 d	3.32 b	4.56 a

Values within columns for each scion/rootstock combination followed by the same letter were not significantly different (*p* > 0.05).

**Table 2 plants-12-04166-t002:** Differentially Expressed Genes (DEGs) of interest between ‘Isabelona’/‘GN-8’ and ‘Isabelona’/Garnem^®^.

logFC	*P. dulcis* ID	Gene	GO Term	Biological Process
1.392	Prudul26A014401	*AGL62*	GO:2000012	regulation of auxin polar transport
1.095	Prudul26A014156	*ATX1*	GO:0009737	response to abscisic acid
1.509	Prudul26A001827	*BPG2*	GO:0009742	Brassinosteroid-mediated signaling pathway
−1.803	Prudul26A026577	*BRC1*	GO:2000032	regulation of secondary shoot formation
0.786	Prudul26A031863	*BZR1*	GO:0009742	Brassinosteroid-mediated signaling pathway
−1.060	Prudul26A020544	*COL2*	GO:2000028	regulation of photoperiodism, flowering
−1.319	Prudul26A006124	*DRM1*	GO:0006346	DNA methylation-dependent heterochromatin formation
−0.796	Prudul26A027812	*DWF4*	GO:0009741	response to brassinosteroid
3.795	Prudul26A014067	*ERF106*	GO:0009873	ethylene-activated signaling pathway
−0.833	Prudul26A003763	*GA3OX1*	GO:0009739	response to gibberellin
1.069	Prudul26A008575	*GASA10*	GO:0009740	gibberellic acid-mediated signaling pathway
−1.422	Prudul26A017626	*GH3.6*	GO:0009733	response to auxin
−0.993	Prudul26A009430	*HVA22D*	GO:0009737	response to abscisic acid
−0.850	Prudul26A024452	*IAA4*	GO:0009733	response to auxin
−0.793	Prudul26A000470	*ICK5*	GO:0007049	cell cycle
0.938	Prudul26A032509	*KAO2*	GO:0009686	gibberellin biosynthetic process
−1.225	Prudul26A009950	*KING1*	GO:0042128	nitrate assimilation
−1.390	Prudul26A008149	*LBD38*	GO:0010468	regulation of gene expression
−1.107	Prudul26A011380	*LSB1*	GO:0006865	amino acid transport
0.913	Prudul26A016769	*MES17*	GO:0048367	shoot system development
−1.064	Prudul26A032130	*MYB3*	GO:0009800	cinnamic acid biosynthetic process
−0.953	Prudul26A007232	*NRT1*	GO:0042128	nitrate assimilation
−1.290	Prudul26A015004	*NRT1*	GO:0042128	nitrate assimilation
1.409	Prudul26A017899	*PDX1*	GO:0030154	cell differentiation
−1.687	Prudul26A015061	*PE11*	GO:0042545	cell wall modification
−1.601	Prudul26A031322	*PERK4*	GO:0009738	abscisic acid-activated signaling pathway
0.780	Prudul26A001835	*PME51*	GO:0042545	cell wall modification
1.440	Prudul26A005193	*RALFL34*	GO:0019722	calcium-mediated signaling
1.179	Prudul26A030616	*RAP2.3*	GO:0009873	ethylene-activated signaling pathway
−1.154	Prudul26A000867	*RAP2.7*	GO:0009873	ethylene-activated signaling pathway
−0.967	Prudul26A017144	*RVE1*	GO:0009734	auxin-activated signaling pathway
−1.305	Prudul26A019438	*RVE7*	GO:0007623	circadian rhythm
−1.080	Prudul26A001965	*SGT1*	GO:0009734	auxin-activated signaling pathway
0.763	Prudul26A008234	*SKU5*	GO:0009932	cell tip growth
0.815	Prudul26A029219	*SRT2*	GO:0009873	ethylene-activated signaling pathway
0.801	Prudul26A019903	*SWI3B*	GO:0006338	chromatin remodeling
−0.828	Prudul26A006961	*ZFP7*	GO:0009738	abscisic acid-activated signaling pathway

**Table 3 plants-12-04166-t003:** Differentially Expressed Genes (DEGs) of interest between ‘Lauranne’/‘GN-8’ and ‘Isabelona’/‘GN-8’.

logFC	*P. dulcis* ID	Gene	GO Term	Biological Process
−1.418	Prudul26A015064	*CLV1*	GO:0030154	cell differentiation
−1.732	Prudul26A007496	*DRMH3*	GO:0048364	root development
−0.767	Prudul26A027600	*EF1A*	GO:0030154	cell differentiation
−2.116	Prudul26A020498	*FIP37*	GO:0010073	meristem maintenance
−1.060	Prudul26A010669	*GRP7*	GO:0007623	circadian rhythm
−0.909	Prudul26A029624	*IAMT1*	GO:0009851	auxin biosynthetic process
−1.209	Prudul26A032200	*JAZ3*	GO:2000022	regulation of jasmonic acid mediated signaling pathway
−0.858	Prudul26A031524	*LAX3*	GO:0009733	response to auxin
−2.079	Prudul26A001972	*RD22*	GO:0009651	response to salt stress
−1.578	Prudul26A025497	*TCTP*	GO:0051301	cell division
−1.542	Prudul26A008660	*VIM1*	GO:0051301	cell division

**Table 4 plants-12-04166-t004:** Differentially expressed genes (DEGs) of interest between ‘Lauranne’/Garnem^®^ and ‘Isabelona’/Garnem^®^.

logFC	*P. dulcis* ID	Gene	GO Term	Biological Process
−1.449	Prudul26A026190	*LTP3*	GO:0009737	response to abscisic acid
0.761	Prudul26A003389	*GLR3.7*	GO:0009737	response to abscisic acid
−1.191	Prudul26A024452	*IAA4*	GO:0009733	response to auxin
−1.679	Prudul26A006124	*DRM1*	GO:0006346	DNA methylation-dependent heterochromatin formation
−1.688	Prudul26A025556	*SAUR50*	GO:0009733	response to auxin
−0.818	Prudul26A024388	*ROPGEF1*	GO:2000012	regulation of auxin polar transport
−1.135	Prudul26A031524	*LAX3*	GO:2000012	regulation of auxin polar transport
−0.803	Prudul26A010146	*TINY2*	GO:0009741	response to brassinosteroid
−1.702	Prudul26A030744	*BAS1*	GO:0009741	response to brassinosteroid
1.428	Prudul26A014814	*SHOC1*	GO:0000712	resolution of meiotic recombination intermediates
1.217	Prudul26A029769	*MCM4*	GO:1902969	mitotic DNA replication
1.192	Prudul26A003343	*RCC1*	GO:0051301	cell division
1.121	Prudul26A018495	*RCC1*	GO:0051301	cell division
0.932	Prudul26A013222	*MCM6*	GO:1902969	mitotic DNA replication
0.779	Prudul26A022273	*PUR4*	GO:0006541	glutamine metabolic process
−1.120	Prudul26A008660	*VIM1*	GO:0051301	cell division
1.657	Prudul26A020211	*4CLL6*	GO:0042545	cell wall modification
0.828	Prudul26A010546	*GALT1*	GO:0006486	protein glycosylation
−0.964	Prudul26A003244	*CYT1*	GO:0030244	cellulose biosynthetic process
−1.009	Prudul26A022981	*NST1*	GO:0009834	plant-type secondary cell wall biogenesis
1.199	Prudul26A005611	*HSFB2B*	GO:0071456	cellular response to hypoxia
0.782	Prudul26A002278	*ZPR1*	GO:0010358	leaf shaping
−0.866	Prudul26A022967	*CAT2*	GO:0009416	response to light stimulus
−1.000	Prudul26A017144	*RVE1*	GO:0009734	auxin-activated signaling pathway
−1.102	Prudul26A014609	*JMJD5*	GO:0007623	circadian rhythm
−1.196	Prudul26A019438	*RVE7*	GO:0007623	circadian rhythm
−1.227	Prudul26A010669	*GRP7*	GO:0007623	circadian rhythm
−3.776	Prudul26A032739	*GRP7*	GO:0007623	circadian rhythm
−1.784	Prudul26A017801	*CKX5*	GO:0009823	cytokinin catabolic process
1.943	Prudul26A021691	*EGY3*	GO:0009651	response to salt stress
1.551	Prudul26A019984	*ERF024*	GO:0009873	ethylene-activated signaling pathway
−0.871	Prudul26A022693	*RCE1*	GO:0009733	response to auxin
2.056	Prudul26A000689	*GA2OX8*	GO:0009686	gibberellin biosynthetic process
−1.423	Prudul26A003763	*GA3OX1*	GO:0009739	response to gibberellin
−1.057	Prudul26A032482	*NAC2*	GO:0009644	response to high light intensity
−1.141	Prudul26A032200	*JAZ3*	GO:2000022	regulation of jasmonic acid mediated signaling pathway
−0.982	Prudul26A012618	*RPT2*	GO:0009638	phototropism
−1.180	Prudul26A007555	*CDF3*	GO:0009908	flower development
−1.273	Prudul26A009950	*KING1*	GO:0042128	nitrate assimilation
1.801	Prudul26A016707	*GI*	GO:0030154	cell differentiation
2.142	Prudul26A024652	*PAT1*	GO:0009640	photomorphogenesis
1.422	Prudul26A014018	*NTL9*	GO:0071470	cellular response to osmotic stress
0.782	Prudul26A016310	*TOE3*	GO:0009873	ethylene-activated signaling pathway
−0.763	Prudul26A027253	*GCR2*	GO:0005975	carbohydrate metabolic process
−0.769	Prudul26A006961	*ZFP7*	GO:0009738	abscisic acid-activated signaling pathway
−0.820	Prudul26A024443	*COL4*	GO:0009909	regulation of flower development
−0.919	Prudul26A002088	*SUI1*	GO:0030154	cell differentiation
−0.994	Prudul26A002057	*LSH3*	GO:0010492	maintenance of shoot apical meristem identity
−1.234	Prudul26A016351	*CDF2*	GO:0009908	flower development
−1.264	Prudul26A025497	*TCTP*	GO:0051301	cell division
−1.291	Prudul26A015967	*SPL9*	GO:0048366	leaf development
−1.586	Prudul26A020498	*FIP37*	GO:0010073	meristem maintenance
−1.971	Prudul26A022418	*MAX1*	GO:0009926	auxin polar transport
0.864	Prudul26A016413	*AGL21*	GO:0048364	root development
−1.010	Prudul26A029350	*JKD*	GO:0048364	root development
−1.082	Prudul26A020939	*AGL79*	GO:0010311	lateral root formation
−1.674	Prudul26A007496	*DRMH3*	GO:0048364	root development
1.705	Prudul26A006492	*SWEET2*	GO:0008643	carbohydrate transport
−0.983	Prudul26A013154	*SWEET2*	GO:0008643	carbohydrate transport

## Data Availability

The datasets generated and/or analyzed during the current study are available in the European Nucleotide Archive (https://www.ebi.ac.uk/ena/browser/home, accessed on 30 July 2021) and are accessible through the accession number PRJEB50411.

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
