# Peer review of "Characterization of Almond Scion/Rootstock Communication in Cultivar and Rootstock Tissues through an RNA-Seq Approach"

_plants, 2023, doi:10.3390/plants12244166_

Round 1

Reviewer 1 Report

Comments and Suggestions for Authors

The study is very interesting and well-conducted. However, I would know why a fold change value between 0.75 and -0.75 was used and not 1 and -1 as in the paper by the same authors "Identification of genes involved in almond scion tree architecture influenced by rootstock genotype using transcriptome analysis."

Author Response

Since not many genes were differentially expressed we decided to lower the cutoff for the fold change, but they are all significant. We stayed within limits of what can be found in the literature .

Reviewer 2 Report

Comments and Suggestions for Authors

Rootstock genotype plays a crucial role in determining various aspects of scion development, and the scion genotype also reciprocally affects rootstock performance. In this manuscript, authors conducted a transcriptomic analysis of the scion/rootstock interaction in young almond trees, with a specific focus on elucidating the scion impact on the rootstock molecular response. Through RNA-Seq analysis, they found the scion genotype has an influence on the rootstock expression profile, including genes related to hormonal regulation, cell division, root development, and light signaling. Overall, this manuscript is interesting and well written, and the results help make correct choice when establishing new almond orchards. I have one minor concern for authors to consider to imporve the manuscript.

1. The gene flow between scion and rootstock may occur frequently, and authors should add the possible effects of gene flow on gene expression profiles to the discussion.

Author Response

When retaking our data, reviewer 2 comments is particularly interesting for the RNA of IAA4 that has previously been described as mobile and is differentially expressed in both scion and rootstock in one condition (Laurane/Garnem vs Isabellona/Garnem). Discussion was modified to include it.  We looked for other genes known to be transferred but they are not differentially expressed.  

Reviewer 3 Report

Comments and Suggestions for Authors

In the manuscript “Characterization of almond scion/rootstock communication in cultivar and rootstock tissues through an RNA-Seq approach”, transcriptome analysis of scion/rootstock interactions in young almond trees revealed that scion genotypes do have an impact on rootstock expression profiles. This intricate interplay between scion and rootstock communication plays a pivotal role in the development of both the scion and rootstock, providing a theoretical basis for the establishment of almond orchards. This manuscript is well done but still has some minor formatting errors.

Some errors or questions lists as follows:

Abstract

what does the almond (Prunus amygdalus (L.) Batsch, syn P. dulcis (Mill.)) planting systems refer to?

Line13: Where does the scion three-dimensional structure fit in?

Results:

Line65please note the pre-paragraph formatting;

Table1; Please align the table datatable 2, 3 are the same.

Author Response

The two comments for the abstract from reviewer 3 are related and we clarified them with some edits in the abstract and the introduction. Controlling the vigor and the 3D architecture allow to have almond tree plantations more densely populated ultimately generating more yield. We added a sentence in the introduction explicitly stating that  and we change the abstract "new almond planting systems" into "new almond intensive planting system"

Format of the paragraph L65 was corrected  

All the tables were left-aligned, also table 4